# The evolution of hematopoietic cells under cancer therapy

Oriol Pich [1], Albert Cortes-Bullich [2], Ferran Muiños [1], Marta Pratcorona[2], Abel Gonzalez-Perez [1,3 ✉] & Nuria Lopez-Bigas [1,3,4 ✉]

Chemotherapies may increase mutagenesis of healthy cells and change the selective pressures in tissues, thus influencing their evolution. However, their contributions to the mutation burden and clonal expansions of healthy somatic tissues are not clear. Here, exploiting the mutational footprint of some chemotherapies, we explore their influence on the evolution of hematopoietic cells. Cells of Acute Myeloid Leukemia (AML) secondary to treatment with platinum-based drugs show the mutational footprint of these drugs, indicating that non-malignant blood cells receive chemotherapy mutations. No trace of the 5-fluorouracil (5FU) mutational signature is found in AMLs secondary to exposure to 5FU, suggesting that cells establishing the leukemia could be quiescent during treatment. Using the platinum-based mutational signature as a barcode, we determine that the clonal expansion originating the secondary AMLs begins after the start of the cytotoxic treatment. Its absence in clonal hematopoiesis cases is consistent with the start of the clonal expansion predating the exposure to platinum-based drugs.

[1] Institute for Research in Biomedicine (IRB Barcelona), The Barcelona Institute of Science and Technology, Barcelona, Spain. [2] Hematology and Hemotherapy Department, Hospital Santa Creu i Sant Pau, Barcelona, Spain. [3] Research Program on Biomedical Informatics, Universitat Pompeu Fabra, Barcelona, Catalonia, Spain. [4] Institució Catalana de Recerca i Estudis Avançats (ICREA), Barcelona, Spain. ✉email: abel.gonzalez@irbbarcelona.org; nuria.lopez@irbbarcelona.org

Somatic tissues evolve as a result of the interplay between genetic variation—contributed by a range of endogenous and external mutational processes—and selective constraints acting at the level of organs or tissues[1–3]. Chemotherapies cause the death of large amounts of cells, thus imposing specific selective constraints on somatic tissues[4]. Certain cells, able to withstand chemotherapies by virtue of certain advantageous mutations or phenotypic characteristics may subsequently expand to replenish the exhausted tissue after the insult is withdrawn. Some widely used chemotherapies, due to their mutagenic mechanism, also contribute to the genetic variation present in exposed tumor cells or cell lines[5–10].

One homeostatic process in which the long-term effects of chemotherapies have been extensively studied is hematopoiesis. It is known, for example, that secondary hematopoietic malignancies, such as acute myeloid leukemia (AML) occur in patients who are exposed to chemotherapies as part of the treatment of a solid malignancy[11,12]. Moreover, clonal hematopoiesis (CH), a condition related to aging across the human population is known to occur more frequently among people previously exposed to chemotherapies[13–19]. CH, in turn, is associated with other health risks, such as subsequent hematopoietic malignancies or increased incidence of cardiovascular disease[13,17,20–22]. The molecular mechanisms underlying the advantage provided by some CH-causing mutations affecting DNA damage and repair genes in the face of certain chemotherapies have been unraveled[16]. However, the timing of the clonal expansion that ultimately causes treatment-related CH or treatment-related AML with respect to the exposure to chemotherapy remains elusive. This is key to understanding whether the cytotoxic agent may be the cause of this clonal expansion or only provides a new evolutionary constraint that favors the expansion of a pre-existing clone.

To date, it is not known whether these chemotherapeutic agents leave their mutational footprint in healthy cells. Furthermore, the immediate effects of the exposure to chemotherapies on the evolution of healthy tissues is not clear. We reasoned that the detection of the mutational footprint of chemotherapies in therapy-related AML (tAML) and CH cases provide a powerful tool to study their influence on such conditions. We first set out to determine whether cells that were not malignant at the time of their exposure to chemotherapeutic agents bear their mutational footprint. Furthermore, we used this footprint as a barcode[23] to determine whether the clonal expansion started before or after the beginning of the exposure to the drug. Ultimately, this allowed us to study how chemotherapies interfere with healthy hematopoiesis, causing the emergence of treatment-related AML (tAML) or treatment-related CH (tCH).

## Results

**The role of chemotherapies in the evolution of tAMLs.** We and others have previously observed that certain chemotherapies, through direct DNA damage or interference with the replication machinery leave a mutational footprint in the metastatic tumors of patients exposed to them as part of the cancer treatment[5,7]. We reasoned that such chemotherapies, being systemic, may also be able to leave their mutational footprint in non-malignant cells. However, detecting private chemotherapy mutations—in the absence of any clonal expansion—in non-malignant cells is extremely challenging. We reasoned that secondary neoplasms could give us the opportunity to study this. Secondary neoplasms, such as tAMLs, which appear in some patients following treatment of a primary solid tumor, originate from hematopoietic cells that were non-malignant at the time of exposure (Fig. 1a). It is not clear whether these tAMLs are driven specifically by drug-

related mutations, or if they appear as a consequence of the evolutionary bottleneck chemotherapies posed on hematopoiesis, or as a contribution of both factors.

To explore the role of anti-cancer treatments in the development of tAMLs, we collected a cohort of 30 (3 in-house) whole-genome sequenced tAMLs (Fig. 1b; Fig. S1a; Table S1) and 32 primary AMLs (WGS AML cohort)[24,25]. Overall, no significant differences are appreciable in total somatic SNVs ($p = 0.33$) or indels ($p = 0.1$) burden between tAMLs and primary AMLs, as previously reported[24] (Fig. 1c; Table S1). However, all tAMLs from patients exposed to platinum-based chemotherapies exhibit a mutational footprint associated with these drugs ($N = 8$; Fig. 1d; Fig. S1b; Table S1). This signature is highly similar (cosine similarity 0.94) to a platinum-related signature identified across metastatic tumors from patients exposed to the drug[5,26]. (Heretofore, we refer to this mutational signature as platinum-related.) The contribution of this signature to the mutation burden of platinum-exposed tAMLs causes their total burden to be significantly higher than that of primary AMLs and tAMLs not exposed to platinum-based drugs (Fig. 1e). Moreover, platinum-exposed tAMLs harbor a significantly higher number of double base substitutions than tAMLs of patients exposed to other drugs and primary AMLs, as expected from the mutagenic mechanism of platinum-based drugs[27] (Fig. 1e; Table S1). Despite the profound phenotypic differences that exist between non-malignant hematopoietic cells and solid tumor cells, the number of platinum-related mutations in tAMLs is similar (around 1000) to that detected across metastatic tumors from patients exposed to the same chemotherapies[5] (Fig. 1f). Therefore, the exposure to platinum-based drugs appears to be a sufficient condition for the acquisition of platinum-related mutations in non-malignant hematopoietic cells.

Interestingly, no trace of the 5FU-related mutational footprint appears in any of the three tAML cases from patients exposed to the drug (Fig. S1b). This absence of the 5FU-related mutational profile is not the result of lack of statistical power (Fig. S2a–e). Rather, the footprint is indeed absent from 5FU-exposed tAMLs. Platinum-based drugs directly damage the DNA of cells generating bulky adducts. Conversely, 5FU through the inhibition of thymidylate synthase[28], alters the pool of nucleotides available for DNA synthesis, and as analogous to thymidine it could, hypothetically, be incorporated to the nascent DNA strand. We thus reasoned that cells that divide during the treatment could incorporate 5FU mutations, whereas cells that remain quiescent during the exposure to 5FU—and do not divide until the nucleotide pool has been recovered—will not. The lack of the 5FU mutational footprint would indicate that cells that establish tAMLs could have been quiescent (potentially hematopoietic progenitors) at the time of treatment[29–31].

In summary, we demonstrated that platinum-based therapies leave their characteristic mutational footprint across non-malignant hematopoietic cells upon exposure to them. Intriguingly, the footprint that characterized 5FU appears to be absent from these exposed non-malignant hematopoietic cells, possibly because quiescence is a key mechanism of survival to the chemotherapy exposure.

**Footprint of chemotherapies with differing mutagenic mechanisms.** To investigate further the hypothesis that the differing mechanisms of platinum-based drugs and 5FU/Capecitabine were the reason why we only observed the footprint of the former, we resorted to a cohort of metastatic cancer patients (more than 700) who as part of the treatment of their primary tumor had been exposed to either of these drugs (metastasis cohort[32]; Fig. S3a). In this cohort, the mutational footprint of

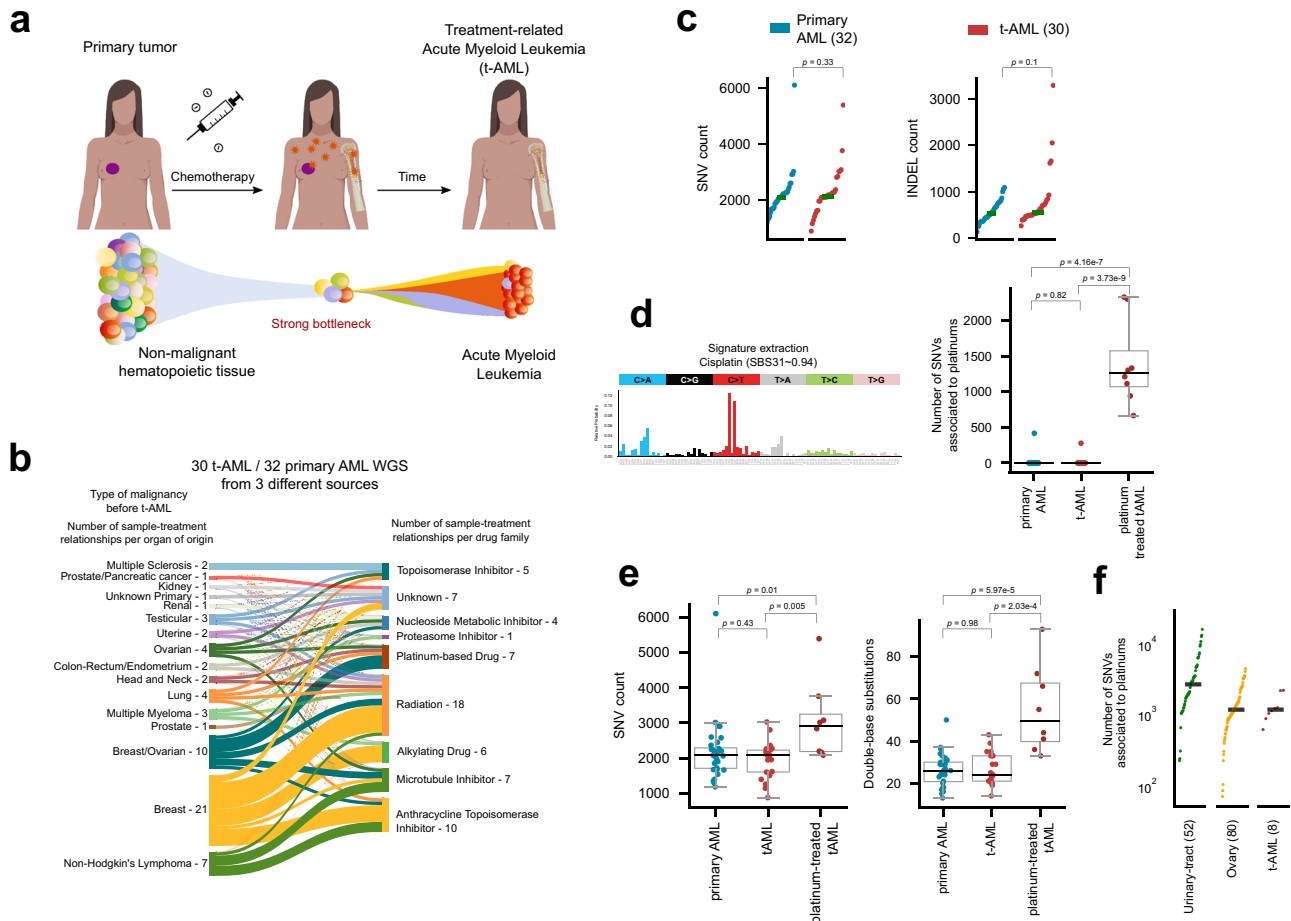

**Fig. 1 The mutational footprint of chemotherapies across treatment-related AMLs. a** Cancer patients (breast cancer as an example in the figure—purple circle) exposed to chemotherapy, which damages cell DNA (orange stars), may develop treatment related acute myeloid leukemia (tAMLs). Non-malignant hematopoietic cells at the time of exposure to chemotherapy are faced with a bottleneck that reduces the population, leading to the development of AML over time. **b** The whole-genome sequence of thirty tAML cases of patients who suffered from a primary solid tumor and were treated with different anticancer drugs (represented in the Sankey diagram) were obtained from three different sources, including three cases sequenced in-house. These were analyzed in combination with 32 cases of primary AMLs (WGS AML cohort). **c** Burden of single nucleotide variants and indels do not significantly differ across cases of the WGS AML cohort (two-tailed Mann–Whitney $p = 0.33$, and 0.1 respectively). **d** Mutational profile of a platinum-related signature active across cases in the WGS AML cohort. All tAML cases from patients exposed to platinum-related drugs exhibit activity of the signature (two-tailed Mann–Whitney $p = 3.37 \times 10^{-9}$, and $4.16 \times 10^{-7}$). As a result of the process of reconstruction of the mutational profile of all samples carried out by the signatures extraction algorithm, one tAML case in a patient not exposed to platinum-based drugs and one primary AML case exhibit a "false" small activity of the platinum-related signature, a phenomenon known as signature bleeding[26]. (See more details in Fig. S1.) **e** Burden of single nucleotide variants (left) and double base substitutions (right) of primary AML, non-platinum-exposed, and platinum-exposed tAML cases in the WGS AML cohort. Comparisons were carried out with a two-tailed Mann-Whitney test. **f** Number of mutations contributed by the platinum-related signature across tAML cases in comparison with that counted across metastatic tumors from several organs of origin. The box in each boxplot delimits the first and third quartiles of the distribution (with a line representing the median); the whiskers delimit the lowest data point above the first quartile minus 1.5 times the interquartile distance and the highest data point below the third quartile plus 1.5 times the interquartile distance. AML acute myeloid leukemia, t-AML treatment-related AML, WGS whole-genome sequencing, SBS single base substitution, SNV single nucleotide variant, INDEL insertion or deletion.

5FU/Capecitabine and platinum-based drugs have been detected by us and others[5,7] in some metastatic tumors (Fig. 2a).

We reasoned that the detection of treatment-related mutations in metastases (upper branch in schematic Fig. 2b) could occur in two different scenarios. In a metastasis seeded before the start of the treatment, the chemotherapy imposes an evolutionary bottleneck that may result in a reduction of the size of the tumor. In the presence of a strong bottleneck the metastasis could be reduced to one or few surviving cells, yielding treatment mutations that can be detected by bulk sequencing. In the second scenario, the treatment predates the seeding of the metastasis. The effect of the bottleneck, in this case, consists in favoring seeding by one or few cells. In either scenario, if the bottleneck is not strong enough, this decreases the likelihood that treatment

mutations appear at levels that are detectable through bulk sequencing (middle branch). In both scenarios, the survival of a single cell that expands to form the biopsied metastasis would result in at least a fraction of the treatment-related mutations being detected as clonal. Differences observed in the fraction of clonal treatment-related mutations across metastatic tumors may therefore also reflect their distinct evolutionary histories (Fig. S3b). Another possibility—with a result indistinguishable from the latter—is that the treatment does not leave a mutational footprint in tumor cells (lower branch).

To further explore what determines that the mutational footprints of platinum-based drugs and 5FU across metastases from exposed donors are detected, we studied whether the metastasis appears in a site proximal or distal of the organ of the

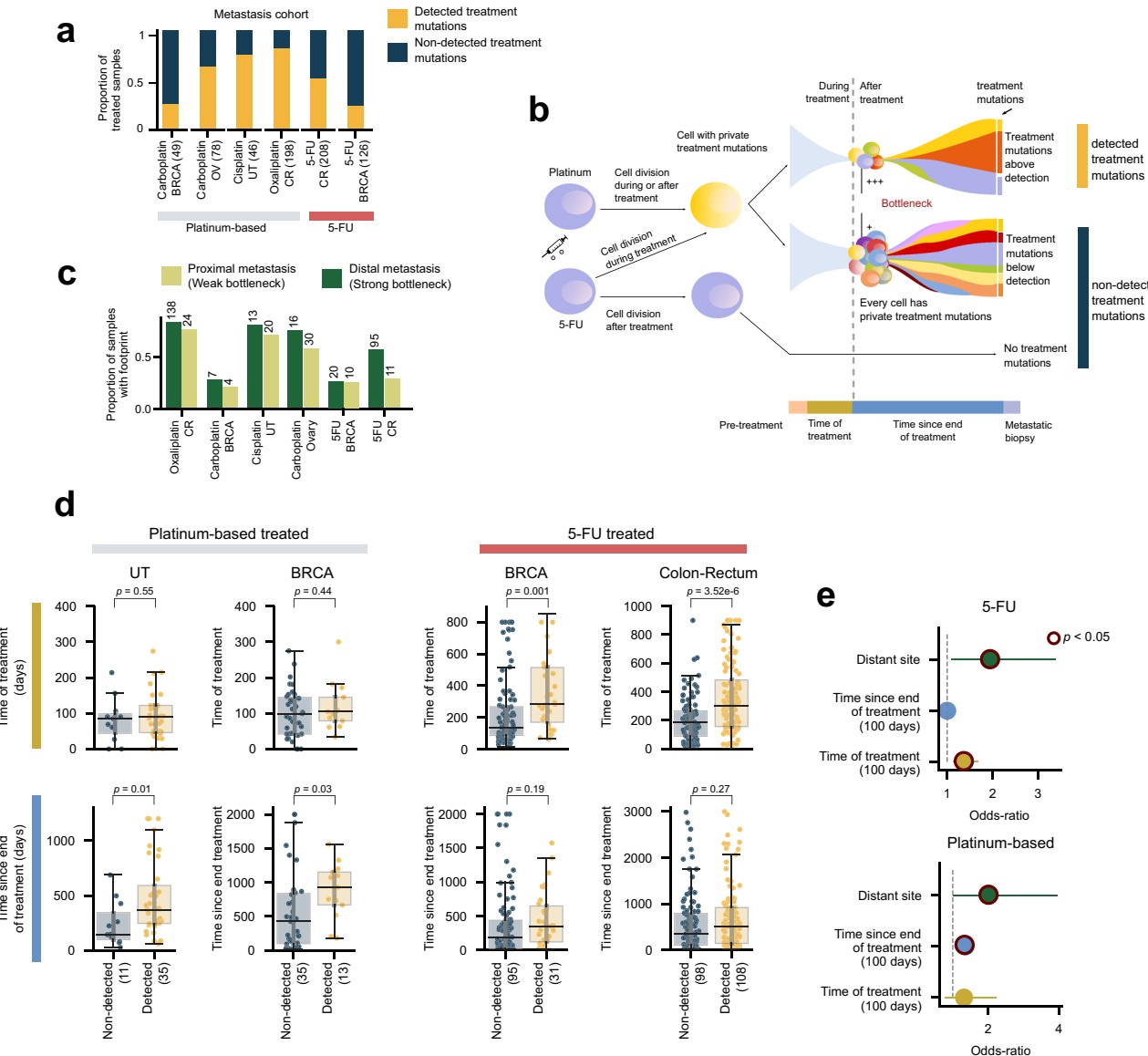

**Fig. 2 Different mutagenic mechanisms of platinum-based drugs and 5FU. a** Proportion of samples from metastatic tumors with different organs of origin taken from donors exposed to platinum-based drugs or 5FU with detectable treatment mutations. **b** Proportion of samples with detectable treatment mutations among distant or proximal metastases. **c** Platinum-based drugs and 5FU contribute mutations in the DNA via two different mechanisms (illustrated by the blue cells in the left panel). While the former creates adducts on the DNA, the latter alters the pool of nucleotides available for DNA replication. Thus, cells exposed to platinum-based drugs will carry mutations of the treatment after DNA replication, irrespective of whether cells are quiescent at the time of exposure or not (top arrow). On the other hand, 5FU-exposed cells will only incorporate mutations if their DNA is replicated while the pool of nucleotides is still distorted by the drug (middle arrow). If it has been restored before DNA replication, no mutations will be incorporated (bottom arrow). Therefore, immediately following exposure, two scenarios are possible: the surviving cells either bear treatment mutations (yellow cell) or not (blue cell). We reasoned that whether the treatment mutations are visible at the time of metastasis depends on the strength of the evolutionary bottleneck facing the primary tumor and its timing with respect to the exposure (see periods of time represented in different colors below figure). A stronger bottleneck during or immediately after treatment will lead to treatment mutations detectable through bulk sequencing (top drawing). On the contrary, a weaker bottleneck, or a bottleneck suffered before treatment (for example, if the metastasis predates the treatment) will lead to treatment mutations below the limit of detection of the bulk sequencing (middle drawing). **d** Distribution of time of treatment (days) and time after treatment (days) of samples of metastatic tumors of different organs of origin with detectable or undetectable treatment mutations taken from patients exposed to platinum-based drugs or 5FU. Groups of metastatic tumors were compared using a one-tailed Mann–Whitney test. The box in each boxplot delimits the first and third quartiles of the distribution (with a line representing the median); the whiskers delimit the lowest data point above the first quartile minus 1.5 times the interquartile distance and the highest data point below the third quartile plus 1.5 times the interquartile distance. **e** Results of logistic regressions showing the influence of different variables on the detection of platinum-based drugs (bottom) or 5FU (top) related mutations across metastatic tumors. Variables that significantly influenced the likelihood of detection are circled. BRCA breast adenocarcinoma, OV ovarian adenocarcinoma, UT urinary tract tumors, CR colorectal adenocarcinoma, 5-FU 5-fluorouracil.

primary tumor and the time elapsed between the start of the treatment and the collection of the metastasis biopsy.

The mutational footprint of both platinum-based drugs and 5FU is detected more often in distal metastases—frequently mono-clonally seeded due to a stronger evolutionary bottleneck[33]—, than in proximal metastases (Fig. 2c, e; Fig. S3c). Tumors with detectable mutational footprint of either treatment exhibited significantly longer time elapsed between the start of the treatment and the metastasis biopsy than those with no detectable footprint (Fig. S3d). Significant differences in the time between the end of treatment and the biopsy (time since the end of treatment) distinguished platinum-exposed tumors, but not 5FU-exposed tumors (Fig. 2d, e; Fig. S3c; Table S2). We reasoned that the longer the time since the end of treatment, the more likely the clonal expansion leading to the metastasis—in any of the two scenarios described above—began after the exposure to the chemotherapy, and thus the higher the odds to detect the footprint.

On the contrary, tumors with detectable footprint of 5FU were exposed to the drug (time of treatment) significantly longer than those without the footprint, but both groups showed no significant differences in the time since the end of treatment (Fig. 2d, e; Table S2). Unlike unrepaired DNA adducts left by platinum-based drugs, which may be directly converted into mutations as DNA replicates, 5FU-generated mutations can only appear if, at the time of replication, the nucleotide pool has not been restored. This is more likely to occur the longer cells are exposed to 5FU, and is probably the reason why the time of treatment is key to observing 5FU mutations in the metastases. This would also explain why there are no significant differences in the time after the treatment for the metastases that show the 5FU footprint and those that do not. The significantly longer time of treatment of tumors with the 5FU footprint may account for the time required for the full clonal expansion.

These results provide clues about the relationship between the strength of the evolutionary bottleneck imposed by the treatment of the primary tumor and the evolution of the metastasis, and of the state of cells during treatment. How this evolutionary bottleneck comes about in the interplay between treatment regimens, doses, the genetic background of the patient, and the genetic and epigenetic makeup of the tumor, among other factors, remains to be studied.

In summary, treatment-related mutations are only detected in a fraction of the tumors in the metastasis cohort, and their level of clonality is variable across metastases. Using the mutational footprint of the chemotherapy as a barcode, we are able to infer that in metastases with high activity of the drug related signature among clonal mutations, the start of the treatment preceded the beginning of the clonal expansion giving rise to the metastasis[23]. The detection of the footprints also provides clues about the state of cells during treatment. Therefore, as stated above, the absence of detectable 5FU-related mutations in tAMLs may be explained by the malignancy being started by hematopoietic cells that were quiescent during their exposure to the drug.

**The effect of chemotherapies on the selective constraints faced by hematopoietic cells**. Next, we used the knowledge leveraged in metastatic tumors to ask whether the clonal expansion of the tAML is prior or posterior to the exposure of patients to platinum-based drugs. We found that across the WGS AML cohort, platinum-related mutations are more active among clonal mutations than those contributed by other mutational processes (Fig. 3a). The presence of that platinum-related footprint among clonal mutations in all platinum-exposed tAMLs is confirmed by a de novo identification of signatures that are active among clonal mutations (Fig. S4a). Furthermore, most of the mutations

contributed by the platinum-based signature are clonal (75%) (Fig. 1d). This is consistent with the treatment predating the clonal expansion of the hematopoietic cells that founded the tAML. The presence of a fraction of subclonal mutations could be explained by mutations that appear during treatment after the initiation of the clonal expansion, or due to lesion segregation[34].

Across both, tAML and primary AML cases, the most prevalent mutational signature is associated with the steady accumulation of mutations in hematopoiesis, or HSC signature[35], (Fig. 3b, c; Fig. S4b; $p = 2.86 \times 10^{-17}$) over time. The linear relationship between the number of mutations contributed by this process of hematopoiesis and age, which has been observed in healthy hematopoietic cells is maintained across primary AMLs, albeit with a slight acceleration (Fig. 3c; $p = 2.34 \times 10^{-11}$). This relationship is maintained when the effect of potential confounding signatures (such as AML1 SBS) and of possible sequencing artifacts is taken into consideration (Fig. S4c). The mutations contributed by other processes active in hematopoietic cells do not accumulate linearly with age (Fig. S4d).

Surprisingly, the linear relationship between the gain of hematopoiesis-related mutations and age at diagnosis is lost across tAMLs, and this appears to be independent of whether or not the chemotherapy leaves a mutational footprint (Fig. 3d, $p = 0.81$). Some chemotherapies, such as 5FU, are known to affect particularly cycling cells[36]. Upon recovery from their effect, the repopulation of the hematopoietic compartment may result in an accelerated HSC mutation rate[37,38]. This would produce—as observed in some tAMLs—a number of hematopoiesis mutations that is higher than that expected given the age of patients (Fig. 3d). Nevertheless, some tAML cases exhibit a mutation burden below the trend of homeostatic accumulation of mutations with age (Fig. 3d), which could be explained by an acceleration not high enough (or lack thereof) to overcome the relatively low burden of the quiescent HSCs. In any event, the loss of linearity between the age of patients and the number of HSC mutations across tAMLs suggests that regardless of their mutagenic effect, chemotherapies alter the developmental dynamics of the hematopoietic compartment[30,38].

Since the hematopoietic cells founding the primary and tAMLs face different selective constraints, we next asked whether mutations in different genes drive both malignancies. To this end, we identified genes under positive selection across 261 whole-exome sequenced AMLs (16 of them secondary to treatment) with paired-normal samples. These are part of a cohort of 608 AML patients[39], 41 of which correspond to tAML cases (WES AML cohort; Fig. S3e). While previous studies have focused on the enumeration of observed mutations in known or suspected cancer genes across tAML cases[40,41], we applied an unbiased approach[42] for a de novo identification of driver genes in the WES AML cohort. We identified signals of positive selection in the mutational pattern of 36 genes (Fig. 3e; Table S3). Although the same set of genes drive both primary and tAML cases, some differences are appreciated in the prevalence of their mutations (Fig. S3f; Table S3), such as an enrichment of TP53 mutations among tAMLs—an effect noted before[24,40,41] (Fig. 3f). Intriguingly, contradicting previous studies involving other cohorts[40,41,43,44], we observed the same for IDH1 mutations. This discrepancy might thus be attributed to specific characteristics of the WES AML cohort. On the contrary, and in agreement with previous observations[40,41], mutations of NPM1 are under-represented across cases of tAML.

**The evolution of treatment-related CH and tAML**. CH is a condition characterized by the presence in the blood of individuals of an expanded HSC clone, driven by advantageous somatic

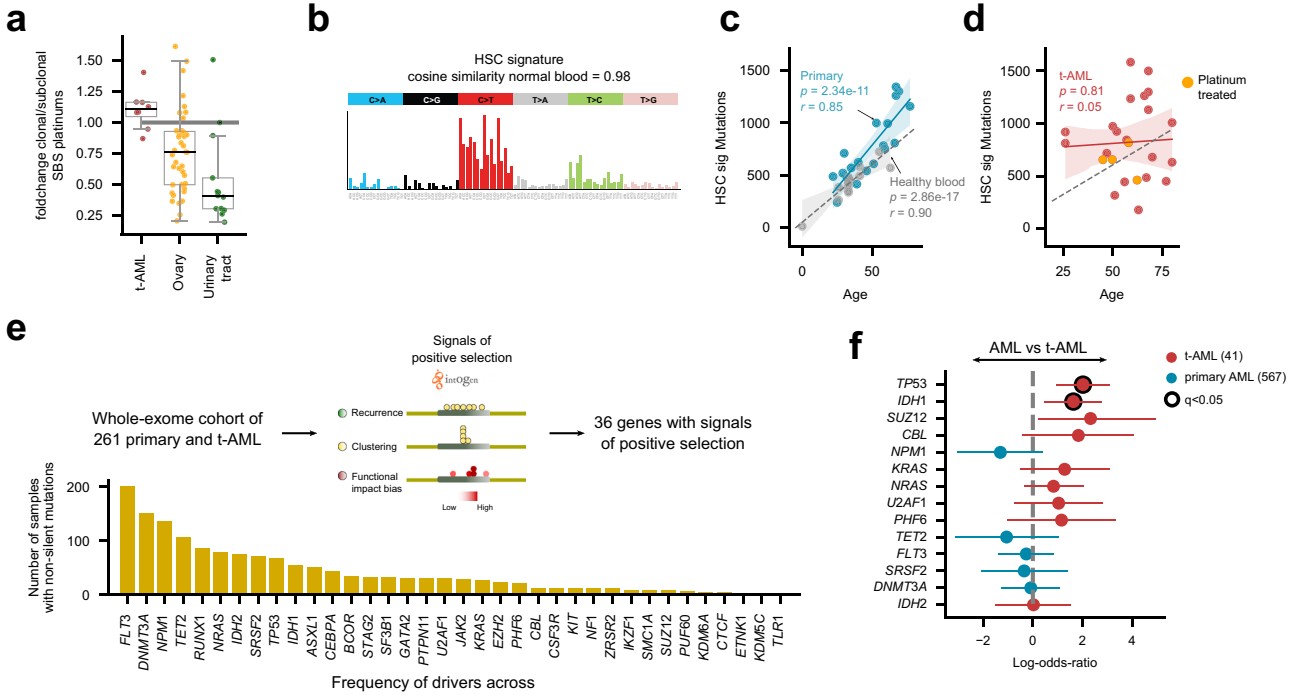

**Fig. 3 The development of treatment-related AMLs. a** Relative activity of platinum-related signature among clonal mutations (with respect to subclonal mutations) across tAMLs and the metastatic tumors of several organs of origin. The box in each boxplot delimits the first and third quartiles of the distribution (with a line representing the median); the whiskers delimit the lowest data point above the first quartile minus 1.5 times the interquartile distance and the highest data point below the third quartile plus 1.5 times the interquartile distance. **b** Mutational profile of the HSC signature active across tAML cases. **c** Linear relationship between the age of patients and the number of HSC mutations of primary AMLs (blue dots). The gray dots correspond to healthy blood donors and thus the regression of their HSC mutation burden to the age represents the normal accumulation of mutations in the process of hematopoiesis. In the figure $r$ represents the two-tailed Pearson's correlation coefficient and $p$, its associated $p$-value. The shaded areas cover the 95% confidence intervals of the corresponding regression lines. **d** A non-significant regression is obtained between the HSC mutation burden and the age of patients with tAML. In the figure $r$ represents the Pearson's correlation coefficient and $p$, its associated $p$-value. The shaded area covers the 95% confidence intervals of the regression line. **e** Genes with detectable signals of positive selection across the subset of 261 AMLs of the WES AML cohort with a healthy control sample. The recurrence of mutation shown by the bars is measured across the entire cohort. **f** Overrepresentation of mutations of different genes across primary or treatment-related AMLs. Significant cases (Benjamini-Hochberg FDR < 0.05) appear circled. The bars represent the 95% confidence intervals of the log-odds ratio. SBS single base substitution, HSC hematopoietic stem cell, sig signature.

mutations[13–17,19,45,46]. In the growth conditions of the healthy bone marrow niche, or faced with particular challenges—such as exposure to cytotoxic therapies—some of these mutations provide an HSC with a growth advantage with respect to its neighbors. We previously detected somatic mutations present in the blood samples of the ~4000 donors of the metastasis cohort using the tumor sample as a reference of the germline genome of the donor[46] (Fig. 4a). The detection of these mutations constitutes evidence of the occurrence of CH in the donors of this cohort. Tracing the signals of positive selection on the mutations observed in genes across donors we have previously obtained a compendium of CH driver genes[46].

Exploiting this compendium of CH drivers, we next asked whether mutations in the same genes provide an advantage to hematopoietic cells faced with chemotherapies to develop treatment-related CH or full-blown tAML. In coherence with observations on the risk that mutations in different genes pose for the progression of age-related CH into AML[47,48], we found mutations of *TP53* (significantly), *IDH2* and several splicing factors (not significantly) overrepresented among tAML cases (Fig. 4b). In addition, mutations of *FLT3*, known to occur late in leukemogenesis[41], were also significantly overrepresented across tAML cases. Conversely, *DNMT3A* mutations appear significantly overrepresented across CH cases, an effect driven mostly by mutations outside the R882 hotspot (Fig. S5a).

Finally, we reasoned that if, as in the case of tAMLs, the treatment predated the start of the clonal expansion, platinum-related mutations would occur at the same clonality as hematopoiesis mutations, as they would have been present in the HSC that founded the CH. We expect a median of 100 platinum-related mutations in each blood sample with detectable CH—i.e., the same number of hematopoiesis mutations, as across tAMLs (Fig. 4c). Therefore, we reasoned that, under these conditions, platinum-related mutations should be detectable across the blood samples of CH cases in the metastasis cohort. However, no platinum mutational footprint (or of any drug) is detected through a mutational signatures extraction approach across the blood samples in the metastasis cohort (Fig. S5b, c). We thus determined the limit of detection of platinum-related mutations across the blood samples with detectable CH in the metastasis cohort through the capability of mSigAct—a method aimed at recognizing the activity of a particular mutational signature—[49] to retrieve the activity of platinum-based drugs generated in a controlled experiment. Briefly, we first generated synthetic samples with analogous distribution of mutation burden and background mutational profile as the blood samples in the metastasis cohort (Supplementary Notes). Then, we injected across these samples increasing numbers of platinum-related mutations and, each time, we asked whether the inclusion of the platinum-based signature significantly improved the

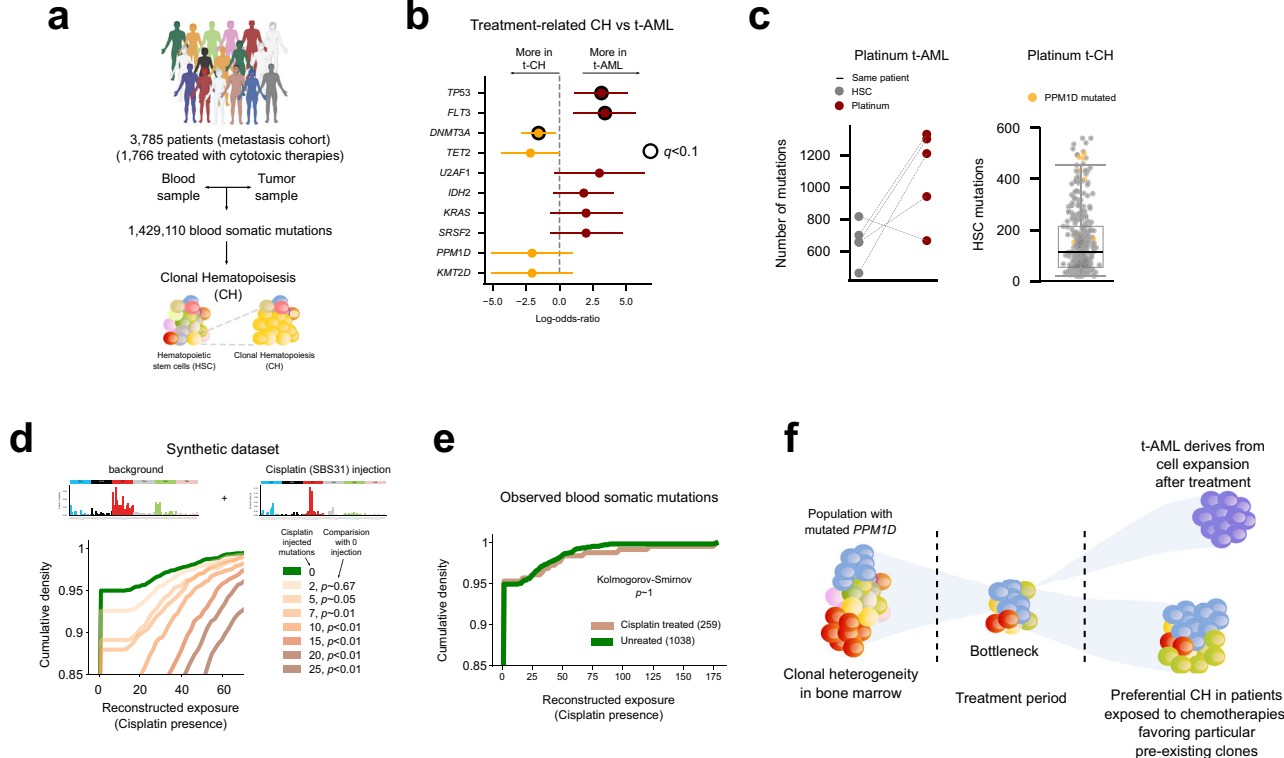

**Fig. 4 The development of clonal hematopoiesis in chemotherapy-exposed donors. a** Somatic mutations across blood samples from 3785 patients in the metastasis cohort were detected using the tumor sample as reference of their germline genome. This identification is possible because a process of clonal hematopoiesis has occurred in these samples, rendering the mutations in the founder HSC detectable through bulk sequencing. **b** Overrepresentation of mutations in different genes across CH or tAML cases. The bars represent the 95% confidence intervals of the log-odds ratio. **c** Left panel: number of mutations contributed by the HSC and the platinum-related signature in tAML cases of the WGS AML cohort. Each patient is represented by a straight line joining two circles that represent the contribution of the HSC (gray) and the platinum-related (red) signatures. Right panel: distribution of the number of HSC mutations identified in blood samples across donors in the metastasis cohort. Samples carrying *PPM1D* mutations are highlighted in yellow. The box in the boxplot delimits the first and third quartiles of the distribution (with a line representing the median); the whiskers delimit the lowest data point above the first quartile minus 1.5 times the interquartile distance and the highest data point below the third quartile plus 1.5 times the interquartile distance. **d** To study the detectability of platinum mutations in blood samples of donors of the metastasis cohort, we generated synthetic samples with the same mutational profile. Then, we injected sets of mutations of different sizes that followed the tri-nucleotide probabilities of the platinum-related signature. We then determined the distribution of the reconstructed exposure to this signature of the injected samples. The cohorts of synthetic samples injected with seven or fewer mutations exhibit a distribution of reconstructed exposure that does not differ significantly (one-tailed empirical *p*-value) from that of the original cohort of synthetic samples. However, the distribution of reconstructed exposure of cohorts of samples injected with 10 or more platinum-related mutations are significantly different from that of the original cohort of synthetic samples. **e** No significant differences (Kolmogorov-Smirnov test) are observed in the distribution of the reconstructed exposure (activity) to the platinum-related signature among platinum exposed and platinum non-exposed donors in the metastasis cohort. **f** Schematic diagram depicting the possible development of CH and tAML upon chemotherapy exposure of hematopoietic cells. CH clonal hematopoiesis, t-CH treatment-related clonal hematopoiesis, SBS31 platinum-related single base substitution signature, HSC hematopoietic stem cell.

reconstruction of the mutational profile of these samples over the signatures included in the background. In this experiment, we assume that platinum-related mutations have the same clonality as HSC mutations, as would be the case if the CH started after the beginning of the treatment, as observed for tAML cases. We ascertained that a significant improvement in the reconstruction of the profile of these synthetic samples ($p < 0.01$) is obtained for 10 or more injected platinum-related mutations (Fig. 4d; Supplementary Notes). Nevertheless, the application of the mSigAct to the blood somatic mutations of the metastasis cohort yielded no trace of the platinum mutational footprint (Fig. 4e; Supplementary Notes), indicating that treatment mutations are below the level of clonality of hematopoiesis mutations. Hence, we concluded that the start of the clonal expansion giving rise to the CH is prior to the beginning of the exposure to platinum-based drugs.

In summary, we posit that small pre-existing clones of hematopoietic cells carrying a mutation that is advantageous to the cells in the presence of platinum-based drugs, such as the *PPM1D* (yellow-colored in Fig. 4c, right) continue their expansion favored against the background of the selective constraint posed by the chemotherapy. Conversely, in the case of tAML, with a stronger bottleneck during or after treatment, a single cell gives rise to the leukemic clone (Fig. 4f).

## Discussion

Sequencing somatic tissues provides a window into their evolution through the interplay of genetic variation and selection. The sources of genetic variation and their change throughout time may be resolved by delineating the mutational processes and clonal sweeps to which a tissue has been exposed[26,50]. The detection of the clonal sweeps punctuating the history of a tissue

informs on the mutations that provide selective advantages to the constraints faced by its cells[51,52]. The link between the evolutionary dynamics of somatic tissues and the emergence of neoplasias has been known for a long time, and as a result, the interplay of variation and selection has been intensively studied in the context of tumorigenesis[1–3]. Nevertheless, studying the evolution of somatic tissues throughout life is also key to understanding aging, neurodegenerative diseases, and certain cardiovascular conditions. In this endeavor, the resources, approaches and methods developed in recent decades to study cancer emergence and evolution may be repurposed to examine other conditions and diseases.

In this article, building upon our previous finding of the mutational footprint of a group of chemotherapies[5], we revealed that mutations contributed by platinum-based drugs appear in cells that were non-malignant at the time of their exposure. Furthermore, the number of treatment mutations that appear in these cells is in the same order of magnitude as those observed in malignant cells after exposure. In other words, at least in the case of platinum-based drugs, which directly damage the DNA[27], the exposure to the drug produces a mutagenic effect of the same magnitude in both tumoral and healthy cells. We propose that mutations caused by these substances appear irrespectively of whether cells are dividing or not at the time of exposure. Other agents require their incorporation into the DNA by polymerases as a necessary step to leave a mutational footprint, such as 5FU/capecitabine and other base analogs[7,53]. Interestingly, their mutations will thus only appear in cells that are not quiescent when the nucleotide pool is altered[29–31,54,55]. This explains why 5FU mutations do not appear in treatment-related AMLs, and only in some healthy colonic crypts of patients exposed to 5FU or capecitabine[56].

One point that remains obscure concerns the potential contribution of platinum-based drugs to the set of driver mutations of treatment-related AMLs. Although mutations in the same genes appear to drive primary and tAMLs, it is still possible that in some cases the mutations driving tAMLs are contributed by the platinum related signature. To solve this issue, more tAML cases from patients exposed to platinum-based drugs need to be sequenced. It is also important to highlight that in this work, we have focused on signatures of single nucleotide variants. It is possible that platinum-based drugs and/or other chemotherapies leave other footprints in the genome in the form of structural variants, some of which might be involved in leukemogenesis.

Exploiting the treatment mutations as a barcode of the clonal expansion of exposed cells, we showed that the original clonal expansion of tAMLs and treatment-related CH differ in their timing with respect to the moment of treatment. While the clonal expansion that founded the tAML is posterior to the moment of treatment, the establishment of the CH clone predates the treatment. In CH, at the time of treatment a variety of small clones of hematopoietic cells already exist. When exposed to platinum-based drugs, HSC clones carrying mutations that hinder the recognition and repair of certain DNA lesions, such as those affecting PPM1D, TP53 or CHEK2 possess an advantage to survive and develop over neighboring clones[15,16,45,46]. Moreover, mutations in different genes are overrepresented across tAML and CH cases. These results suggest that CH and tAML follow different evolutionary pathways. Whether other pre-existing differences at the moment of exposure to the treatment determine one or the other outcome remains to be clarified. The use of treatment mutations as barcodes in the metastatic setting—that track the strength of the bottleneck—may be interesting for clinical applications. Clonal neoantigens are more likely to elicit an antitumor immune response[57,58]. Patients with metastases that underwent stronger evolutionary bottlenecks are expected to carry a higher burden of clonal neoantigens—treatment-related or otherwise—and thus respond better to immune checkpoint inhibitors.

In summary, our results demonstrate the usefulness of certain mutational signatures, such as those associated with the exposure to chemotherapies to study the evolution of somatic tissues. Furthermore, it opens the door to explore some of the longer-term effects that the exposure to such cytotoxic treatments causes in cancer survivors.

## Methods
### Datasets of tumor somatic mutations
*(A) WGS AML cohort.* **Sequencing in-house treatment-related AML samples:** Informed consent was obtained from three secondary AML patients after approval by the Ethics Committee for Clinical Research of Hospital Santa Creu i Sant Pau (January 2017). Within the context of the tAML study described in this paper, the use of these samples was approved by the Ethics Committee for Research of Hospital Clínic de Barcelona (September 2020). DNA was extracted from six samples of three secondary AML patients. Both samples were bone marrow aspirates corresponding to the diagnosis of the AML and the remission. The latter was used as control for the germline genome of the patients. The short-insert paired-end libraries for the whole genome sequencing were prepared with KAPA HyperPrep kit (Roche Kapa Biosystems) with some modifications. In short, in function of available material 0.1–1.0 μg of genomic DNA was sheared on a Covaris™ LE220-Plus (Covaris). The fragmented DNA was further size-selected for the fragment size of 220–550 bp with Agencourt AMPure XP beads (Agencourt, Beckman Coulter). The size selected genomic DNA fragments were end-repaired, adenylated and ligated to Illumina platform compatible adaptors with Unique Dual matched indexes or Unique Dual indexes with unique molecular identifiers (Integrated DNA Technologies). The libraries were quality controlled on an Agilent 2100 Bioanalyzer with the DNA 7500 assay for size and the concentration was estimated using quantitative PCR with the KAPA Library Quantification Kit Illumina® Platforms (Roche Kapa Biosystems). To obtain a sufficient amount of libraries for sequencing it was necessary for the low input libraries (0.1–0.2 μg) to amplify the ligation product with 5 PCR cycles using 2× KAPA-HiFi HS Ready Mix and 10X KAPA primer mix (Roche Kapa Biosystems). The libraries were sequenced on HiSeq 4000 or NovaSeq 6000 (Illumina) with a paired-end read length of 2 × 151 bp. Image analysis, base calling, and quality scoring of the run were processed using the manufacturer's software Real-Time Analysis (HiSeq 4000 RTA 2.7.7 or NovaSeq 6000 RTA 3.3.3). Mean read depth of the tAML samples achieved in the sequencing was 103.

**Calling somatic mutations**: We obtained 57 whole-genome sequenced AML cases from dbGAP phs000159[24]. Two extra samples of tAML from platinum-treated patients were obtained from EGAD00001005028[25]. (Informed consent from the patients was obtained by the original projects that sequenced the samples.) Primary and tAML samples were sequenced at 31 and 35 mean read depth, respectively. The data for each case comprised paired bone marrow samples at the time of diagnosis and remission. The latter was used as control of the germline genome of each patient (Table S1). The cram files deposited were reverted to fastqs using bamtofastq[59]. Then, the fastq files from phs000159, EGAD00001005028, and the inhouse cohort were processed in a uniform manner using the sarek[60] pipeline implemented within nextflow-core (nextflow version 19.10)[61]. Briefly, the pipeline aligns the fastqs to GRCh38 using bwa-mem[62], and implements GATK[63] best practices to mark duplicates and base recalibration, and lastly somatic variant calling. Variant calling of both single nucleotide variants and short insertions and deletions was performed using Strelka2[64]. Only variants labeled as PASS by the pipeline were kept. Variants within regions of low mappability or low complexity[65,66] were excluded from downstream analyses. All somatic mutations were annotated with VEP[67] (version 92).

*(B) WES AML cohort.* Within the beatAML cohort[39], a bone marrow sample and a paired skin sample were taken from 261 AML patients. The somatic mutations identified in these AML cases were downloaded from the Genomics Data Commons[68] (GDC) repository provided by the authors of the original paper reporting the sequencing and variant calling of these patients. Of note, the original Varscan2[69] variant calling was used, and mutations that coincided with variants present in gnomAD[70] v2.1 at allele frequency greater than 0.005 were removed. Mutations identified in the 347 remaining patients without a paired normal sample, as well as the clinical information of all patients in the cohort, were downloaded from the cbioportal repository[71]. While the somatic mutations across the 261 cases with paired samples (deemed more reliably true somatic calls) were employed in the driver discovery (see below), mutations identified in AML driver genes across the 608 cases were subsequently taken into account.

*(C) Metastasis cohort.* Single base substitutions (SBS) identified across the whole-genome of metastases from 729 Breast, 537 Colon-Rectum, 154 Urinary-Tract, and 155 Ovary primary tumors were retrieved from the Hartwig Medical Foundation[32] (HMF) (DR-110). Somatic mutations (called by HMF) were processed as explained

in our previous publication[5]. Briefly, We kept only mutations labeled as PASS by the calling pipeline and filtered out mutations in poorly mappable and low-complexity regions of the genome, as explained above. Clinical data of the donors of these metastatic samples, obtained from HMF were processed, so that treatment regimen were mapped to their unitary drugs and manually assigned drugs administered to 58 different FDA drug categories (https://www.accessdata.fda.gov/cder/ndctext.zip). The start and end dates of treatments and the date of metastasis biopsy were used to calculate the time of treatment and the time between the end of treatment and the biopsy.

*(D) Blood samples of donors of the metastasis cohort.* The WGS of both the tumoral and blood control of the metastatic samples were obtained from the HMF repository for downstream use in the identification of blood somatic mutations. The identification of the somatic mutations in the blood samples of this cohort is explained in Pich et al.[46]. Briefly, the variant calling was carried out using the Google Cloud Platform (metastasis cohort). The matched blood and tumoral BAM files—masked and deduplicated using GATK[63]—were aligned to the human reference genome. The variant calling was carried out with Strelka2[64] employing the blood sample as tumoral input and the tumor sample as control (reverse calling). One metastatic sample was selected in the case of donors with more than one. Only variants with two or more supporting reads matching the caller PASS filter and with $VAF < 0.5$ were kept, and mutations in lowly mappable regions were excluded. Variants observed at a greater frequency than known CH driver mutations (see Pich et al.[46]) across a Panel of Normals (obtained from HMF) or gnomAD[70] v2.1 were removed, as were common SNPs and mutations within segmental duplications, simple repeats, and masked regions. Finally, samples with the mutation count in the 97.5 percentile of the mutation burden across the cohort were deemed unreliable and excluded. Variants remaining after these filters were deemed blood somatic mutations across the donors of metastatic tumors.

*(E) Healthy blood samples.* Whole-genome somatic variants of 23 healthy blood samples were obtained from Osorio et al.[35].

**Mutational signatures extraction**. Mutational signature extraction in the leukemias in WGS AML cohort, the tumors in the metastasis cohort, and the healthy blood samples in the metastasis cohort was performed using a non-negative matrix factorization approach[72,73]. We employed the SigProfilerJulia (bitbucket.org/bbglab/sigprofilerjulia) implementation built in our lab of the algorithm developed by Alexandrov et al.[5,72]. The resulting signatures were then compared to the PCAWG COSMIC V3[26] set using the cosine similarity measure. The Hematopoietic Stem Cell Signature (HSC Sig[35]) was computed as the average of the number of mutations observed across the 23 healthy blood samples in each of the 96 tri-nucleotide channels and normalizing them by the total number of mutations observed.

To test for the activity of a mutational signature in a specific sample we used the mSigAct method[49]. Briefly, given a set of signatures bound to explain the mutational catalogue of the samples (background signatures), this method tests whether an additionally given signature (foreign signature) does improve the sample catalogue reconstruction significantly. The method models the mutation count data as being negative binomial distributed and conducts a likelihood ratio test comparing the likelihood of the observed catalogue under two competitive models: with/without the foreign signature. The method returns for each sample the fitting exposures attributed to each signature (both background and foreign) alongside the significance (*p*-value) yielded by the likelihood ratio test.

In the case of the metastatic samples the footprint was deemed as detectable when both SigProfilerJulia and mSigAct report that the treatment signature was active in the treated sample. The set of signatures deemed relevant according to PCAWG[26] in the respective tumor type cohort was used as a background.

The mutational profile of the 30 tAMLs in the WGS AML cohort was reconstructed with a linear combination of the mutational signatures extracted from a large pan-cancer cohort[26] (see above) and the 5FU-related signature identified in a previous article[5]. The deconstructSigs[74] and the mSigAct[49] methods were employed in the reconstruction. As output of the reconstruction using deconstructSigs we registered the activity (i.e., number of mutations) attributed to the therapy-related signature. In the case of the mSigAct, the significance attributed to the improvement of the reconstruction triggered by the inclusion of the therapy-related signature (either platinum or 5FU) and its attributed activity were measured.

Thirty-five de novo signature extractions were carried out (as explained above) from all possible combinations of three annotated platinum-exposed tAMLs (out of seven) and all the other AMLs in the WGS AML cohort. Each time, the mutational profile of the signature most similar to SBS31 (and its cosine similarity) was computed. Both were averaged across the 35 iterations.

Three synthetic tAML samples were generated using a combination of the HSC signature and noise. Then, 279 (sd 71) mutations were injected following the probabilities derived from the mutational profile of the 5FU-related signature (SBS17b). A de novo signature extraction pooling these three synthetic samples and the WGS AML cohort was carried out, and a signature with high cosine similarity to the 5FU-related signature was detected.

**Driver discovery**. The discovery of genes with signals of positive selection in their mutational pattern across the WES AML cohort was carried out using the IntOGen pipeline[42]. Briefly, the IntOGen pipeline integrates seven complementary methods to identify signals of positive selection in the mutational pattern of genes. The IntOGen pipeline first pre-processes the somatic mutations in a cohort of tumors to filter out hypermutators, map all mutations to the GRCh38 assembly of the human genome and retrieve information necessary for the operation of the seven driver detection methods. Then, the methods are executed and their outputs combined using a weighted voting approach in which the weights are adjusted depending on the credibility awarded to each method. Finally, in a post-processing step, spurious genes that result from known artifacts are automatically filtered out. The version of the pipeline used in this study to identify genes under positive selection across the WES AML cohort and the blood samples of the metastasis cohort is described at length at www.intogen.org/faq and in Martinez-Jimenez et al.[42].

To discover drivers across the WES AML cohort, we selected 261 patients with a matched healthy sample, whose somatic mutations were thus considered more reliable. We ran the IntOGen pipeline on the mutations of these samples. Mutations in these or other genes known to drive AMLs[75] were selected across the 608 patients of the cohort to show in the heatmap of AML drivers. In addition, two tAML cases in the cohort carried the MLL driver translocation.

**Clonal and subclonal mutations**. We used the MutationTime.R package[50] (metastasis cohort) and the approach developed by McGranahan et al.[76] (tAMLs) to classify SBS in a tumor as clonal or subclonal. Then, we associated each mutation uniquely with a mutational signature using a maximum likelihood approach[5,77], and we computed the proportion of clonal mutations amongst all platinum-associated mutations across metastatic tumors.

Subsequently, we computed the fold change between the relative proportions of clonal and subclonal mutations associated to the platinum-related signature.

**Logistic regressions**. Multivariate logistic regression analysis was carried out to explore the association between a set of clinical factors and the ability to identify a specific mutational signal associated with chemotherapy. The factors considered were: time of treatment, time since the end of the treatment, specific tumor type, and whether the metastatic site was considered distal. We considered any metastasis in the same organ as the primary tumor, in a lymph node, or in the omentum or peritoneum (in the case of abdominal primary sites) as proximal to the primary tumor. Metastases in other sites were considered distal. Only tumor types with more than 10 samples were considered for this analysis.

Different multivariate logistic models were considered, each reflecting the sought effect and interactions between the covariates in plausible ways. Model selection was then performed by computing the Bayesian Information Criterion (BIC) and by cross-validation (30% test size, $N = 100$ randomizations) resulting in an average area under the ROC curve (auROC). Model selection gave precedence to models with low BIC and high auROC (Table S2). In both platinum and 5FU treatments, the best model has the same form.

**Compendium of CH drivers**. A list of 64 genes bearing mutations with signals of positive selection in blood samples was obtained from Pich et al.[46].

**Detection of chemotherapy-associated signatures in blood somatic mutations in the metastasis cohort**. We laid out a computational analysis to ascertain the presence of a chemotherapy-induced signature in the blood samples of the metastasis cohort. Given that the low mutation count yielded by the reverse somatic mutation calling precludes a straightforward interpretation sample by sample (high false discovery rate and low statistical power) we proceeded by comparing the distribution of actual reconstructed exposures (inferred from the samples of treated and untreated donors, respectively) against the distribution of exposures reconstructed from synthetic catalogues where the foreign signature has been injected at known levels of exposure.

Given each of 4 possible foreign signatures of interest (associated with platinum and 5FU)[5], we generated synthetic mutational catalogues with/without mutations contributed by the foreign signature. Next, we interrogated the observed and simulated catalogues with a state-of-the-art signature detection method that yields a reconstructed exposure of the foreign signature and a significance level sample by sample[49]. With these outputs we could assess the distributions arising from both observed and synthetic catalogues, thereby assessing the activity of specific mutational processes in a cohort of samples and empirically validating the sensitivity and specificity of the approach. We provide a full description of the methodology in the Supplementary Notes.

**Reporting summary**. Further information on research design is available in the Nature Research Reporting Summary linked to this article.

## Code availability

All analyses described in the paper were implemented in Python and R. All the code needed to reproduce all analyses in the paper is available at https://bitbucket.org/bbglab/evolution_hemato_therapy.

## Data availability

The data employed in the paper is available through different sources. Whole-genome sequences of AML in-house samples that were generated for this study are available on the European Genome-Phenome Archive (EGA) under accession number EGAS00001005234. Whole-genome sequences of previously published samples in the WGS AML cohort are available through dbGAP under accession number phs000159 and on EGA under accession number EGAD00001005028. Access to these datasets may be requested via the European Genome-Phenome Archive or the Database of Genotypes and Phenotypes, and is granted by the respective Data Access Committees. Somatic mutations of samples in the WES AML cohort are available through the GDC repository provided by the authors in the original publication of the beat AML cohort ([https://pubmed.ncbi.nlm.nih.gov/30333627/]) and through the cbioportal under accession number aml_ohsu_2018. Whole-genome sequences of tumor and blood samples in the metastasis cohort are available from the Hartwig Medical Foundation for academic research upon request ([https://www.hartwigmedicalfoundation.nl/en]). (Detailed instructions to access the data of the Hartwig Medical Foundation may be found at https://hartwigmedical.github.) Blood somatic mutations identified in this cohort are also available from the Hartwig Medical Foundation for academic research upon request ([https://hartwigmedical.github.io/documentation/data-access-request-application.html]), due to the extreme difficulty to fully anonymize the data. A detailed description of this dataset of healthy blood somatic mutations identified across patients with metastatic tumors included in the Hartwig Medical Foundation cohort appears at https://doi.org/10.1101/2020.10.22.350140.

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

## Acknowledgements

N.L.-B. acknowledges funding from the European Research Council (consolidator grant 682398) and ERDF/Spanish Ministry of Science, Innovation and Universities—Spanish State Research Agency/DamReMap Project (RTI2018-094095-B-I00) and Asociación Española Contra el Cáncer (AECC) (GC16173697BIGA). IRB Barcelona is a recipient of a Severo Ochoa Centre of Excellence Award from the Spanish Ministry of Economy and Competitiveness (MINECO; Government of Spain) and is supported by CERCA (Generalitat de Catalunya). O.P. is the recipient of a BIST PhD fellowship supported by the Secretariat for Universities and Research of the Ministry of Business and Knowledge of the Government of Catalonia, and the Barcelona Institute of Science and Technology (BIST). The top part of Fig. 1a was created using biorender.org. This publication and the underlying research are partly facilitated by Hartwig Medical Foundation and the Center for Personalized Cancer Treatment (CPCT) which have generated, analyzed, and made available data for this research. This work also benefited from DNA whole-exome sequencing data of the BEAT AML cohort, generated within the BEAT AML clinical trial. We also acknowledge the support provided by Miguel L. Grau and David Martinez to reviewing and maintaining the code developed within the study.

## Author contributions

O.P., A.G.-P., and N.L.-B. designed the project. O.P. carried out all the analyses and prepared the figures. A.C.-B. and M.P. collected the samples and participated in the development of the project. F.M. contributed to the statistical analyses, conducted the analysis of mutational activity with low-count CH data, and wrote the Supplementary Notes. N.L.-B. and A.G.-P drafted the manuscript. O.P., A.G.-P., and N.L.-B. edited the manuscript. A.G.-P. and N.L.-B. supervised the project.

## Competing interests

The authors declare no competing interests.
