## [Peer Review File · Nature Communications]

REVIEWERS' COMMENTS

Reviewer #1 (Remarks to the Author):

The authors have provided an extensive, thorough, and convincing response to all reviewer concerns. This is an important contribution, and I have no further comments.

Reviewer #2 (Remarks to the Author):

I previously expressed a favorable opinion of this paper, except for a few minor comments that have been fully addressed in the new version.

I have only two suggestions, mostly aimed to eventually help future readers:

1) I would include the explanation of inter-bleeding of platinum signature for the two non-exposed cases in Figure 1d legend. This might help readers to understand the concept of “bleeding” and false positive in the mutational signature analysis.

2) I would include either in the manuscript or in Figure S1 legend, the explanation about the patient with testicular cancer without platinum exposure in his/her history, but with clear evidence of platinum mutational signature. I fully agree with Author's interpretation provided in the rebuttal, but I think it would be helpful to include a similar explanation in the manuscript/supplementary. Eventually, I would consider showing the patient's DBS profile, with the platinum associated DBS5 signature.

Francesco Maura, MD, Sylvester Comprehensive Cancer Center - University of Miami

Reviewer #4 (Remarks to the Author):

I really enjoyed reading the Pich et al manuscript and believe the manuscript has strengthened based on the prior reviewer comments and responses to them. I think the reviewers have responded to the comments put forward by reviewer #3 sufficiently and I would recommend publishing the manuscript.

Very minor comments:

* on page3: "However, all tAMLs from patients exposed to platinum-based chemotherapies exhibit a mutational footprint associated with these drugs (N=8; Fig. 1d; Fig. S1b; Table S1)" -> I believe the Fig1B shows N=7 for platinum based drug treatment.

* on page 4: "To investigate further the hypothesis that the differing mechanisms of platinum-based drugs and 5FU/Capecitabine,..." this part of the sentence seems incomplete?

* on page 6: "Surprisingly, the linear relationship between the gain of hematopoiesis-related mutations and age is lost in tAMLs, and apparently, this is independent of whether or not the chemotherapy leaves a mutational footprint (Fig. 3d, $p=0.81$)" -> it is not clear if this analysis accounts for the time between administration of therapy and sampling of blood with tAML.

Reviewer #5 (Remarks to the Author):

Pich et al define mutational footprints in the genomes of AMLs and CH from patients treated with chemotherapeutics for solid cancers, providing new insight into the timing of bottlenecks and clonal expansions. The authors have thoroughly addressed reviewer concerns, by performing new analyses and simulations. Regarding the concerns of Reviewer 3 that the presence of clonal mutations with a platinum signature cannot be used to indicate the timing of the last bottleneck (or last common ancestor), the rebuttal of the authors is convincing and in my opinion their data do support their conclusions in this regard. That the majority of platinum related mutations are clonal is best explained by a selective sweep that occurred after drug exposure. They provide additional analyses restricted to clonal mutations to bolster this conclusion. I would also agree with the authors that "use of treatment mutations as barcodes to determine whether the clonal expansion occurs before or after the start of the exposure to the treatment is ... novel", and adds significantly to our understanding of clonal dynamics in CH and AML evolution. They also use simulations to show that if the 5FU signature were present in the 3 analyzed tAMLs that they would have been able to detect it, addressing this concern raised by Reviewers 1 and 3. They also now show that the 5FU signature is detected in a subset of metastases dependent on certain variables (detailed below). Finally, issues of robustness of data are well addressed through their statistical analyses.

In all, this manuscript provides novel insight into malignant evolution:

1) tAMLs from patients exposed to platinum-based chemotherapies (but not 5FU) exhibit a mutational footprint associated with these drugs. They conclude that "lack of the 5FU mutational footprint would indicate that cells that establish tAMLs were quiescent", which is substantiated by simulations and via the detection of the 5FU signature in metastases from solid cancers.

2) They also analyzed chemo-footprints in the metastatic lesions from this cohort. It is interesting that treatment related mutations are only detected in metastases, and detection is more frequent in metastases that are distal to the primary tumor and with greater time from end of platinum therapy, both of which are plausibly proposed to reflect the increased odds of a bottleneck allowing fixation of particular mutations in one or a few cells (and thus mutations are detectable by WGS). Hence, they are able to infer the timing of the start of clonal outgrowth of the metastasis relative to treatment (if after treatment, therapy related mutations are more likely to be clonal).

3) Further analyses of the AML data show that the treatment likely predated the clonal expansion of the hematopoietic progenitor cells that founded the tAML. The loss of the linear relationship

between the gain of HSC signature mutations (aging related) and age in tAMLs is very interesting, and could indeed reflect the compensatory proliferation during repopulation post-chemo. They used an unbiased method (IntOGen pipeline, based on recurrence, clustering and functional impact) to identify driver mutations in tAMLs, and observed increased driver mutations in TP53 as observed previously. They also observed more IDH1 mutations (in contrast to previous studies), and underrepresentation of NPM1 mutations. While drivers identified are similar to those of other studies, the leveraging of the IntOGen pipeline adds robustness to these identifications. It would be interesting to know whether driver mutations identified in tAMLs or therapy-associated CH display the chemo signature. However, they would likely need to analyze (many) additional tAMLs to determine this. The available sample size is too small.

4) They characterized CH in the blood of the same cohort of patients with metastatic cancers. They observe significant overrepresentation of TP53 (consistent with these mutations conferring a higher risk of AML progress observed by others) and FLT3 (consistent with the occurrence of these mutations late in leukemia evolution), and underrepresentation of DNMT3a mutations. Interestingly, they were unable to detect the platinum mutation signature in CH from treated patients, although they were well powered to do so (based on new simulations). Thus, CH expansion likely occurred prior to the start of platinum treatment, with treatment favoring further expansion. Importantly, their demonstration that tAMLs do display such signatures indicates passage through a bottleneck during or after treatment (likely driven by additional driver mutations).

5) In all, leveraging chemo-related and other mutational signatures, this manuscript provides exciting new insight into clonal dynamics, particularly during CH and AML evolution, in response to chemo treatment. The new insight into the timing of bottlenecks and selective expansions enriches our understanding of malignant evolution in response to therapies.

Minor points:

1) The legend to Figure 4 is jumbled.

2) In the Discussion, I have trouble following this sentence: "This exposure appears as determinant, regardless of the phenotypic state of the exposed cell, to observe mutations related to treatments that directly damage the DNA, such as platinum-based drugs 27."

Signed: James DeGregori

REVIEWERS' COMMENTS

Reviewer #1 (Remarks to the Author):

The authors have provided an extensive, thorough, and convincing response to all reviewer concerns. This is an important contribution, and I have no further comments.

We thank the reviewer for their appreciation of our responses and helpful comments.

Reviewer #2 (Remarks to the Author):

I previously expressed a favorable opinion of this paper, except for a few minor comments that have been fully addressed in the new version.

We thank the reviewer for their support and helpful comments throughout the entire review process.

I have only two suggestions, mostly aimed to eventually help future readers:

1) I would include the explanation of inter-bleeding of platinum signature for the two non-exposed cases in Figure 1d legend. This might help readers to understand the concept of “bleeding” and false positive in the mutational signature analysis.

Following the reviewer’s suggestion, we have added the following explanation to the legend of Figure 1d.

As a result of the process of reconstruction of the mutational profile of all samples carried out by the signatures extraction algorithm, one tAML case in a patient not exposed to platinum-based drugs and one primary AML case (incorrectly) exhibit a small activity of the platinum-related signature, a phenomenon known as signature “bleeding”²⁶.

2) I would include either in the manuscript or in Figure S1 legend, the explanation about the patient with testicular cancer without platinum exposure in his/her history, but with clear evidence of platinum mutational signature. I fully agree with Author's interpretation provided in the rebuttal, but I think it would be helpful to include a similar explanation in the manuscript/supplementary. Eventually, I would consider showing the patient's DBS profile, with the platinum associated DBS5 signature.

Following the reviewer’s suggestion, we have extended the explanation provided in the legend of the Extended Data Figure 1.

A tAML case with activity of the platinum-related signature (and the highest number of double-base substitutions in the entire cohort) from a patient who according to their clinical data was not exposed to any platinum-based drug is marked with an asterisk. We reasoned that,

despite the absence of record, a platinum-based drug was part of the treatment of this patient for their primary tumor.

Francesco Maura, MD, Sylvester Comprehensive Cancer Center - University of Miami

Reviewer #4 (Remarks to the Author):

I really enjoyed reading the Pich et al manuscript and believe the manuscript has strengthened based on the prior reviewer comments and responses to them. I think the reviewers have responded to the comments put forward by reviewer #3 sufficiently and I would recommend publishing the manuscript.

We thank the reviewer for their appreciation of our work.

Very minor comments:

* on page3: "However, all tAMLs from patients exposed to platinum-based chemotherapies exhibit a mutational footprint associated with these drugs (N=8; Fig. 1d; Fig. S1b; Table S1)" -> I believe the Fig1B shows N=7 for platinum based drug treatment.

The reviewer is absolutely right. According to the clinical data shown in Figure 1b, only in 7 tAML cases, the patients received platinum-based drugs as part of the treatment of their primary tumors. However, as explained in the legend of Figure 1 and Extended Data Figure 1 (as explained above), in an eighth tAML case we found all the hallmarks of exposure to these drugs (i.e., high activity of the signature and high number of double base substitutions), despite the lack of record to this effect in their clinical history.

* on page 4: "To investigate further the hypothesis that the differing mechanisms of platinum-based drugs and 5FU/Capecitabine,..." this part of the sentence seems incomplete?

We thank the reviewer for spotting this typo. We have corrected it.

* on page 6: "Surprisingly, the linear relationship between the gain of hematopoiesis-related mutations and age is lost in tAMLs, and apparently, this is independent of whether or not the chemotherapy leaves a mutational footprint (Fig. 3d, $p=0.81$)" -> it is not clear if this analysis accounts for the time between administration of therapy and sampling of blood with tAML.

In this analysis, we use the age of the patients at the time of extraction of the bone marrow sample, which corresponds with the diagnosis of the AML, either primary or therapy-related.

We have added the phrase "at diagnosis" to clarify this point in the pertinent section of results.

Reviewer #5 (Remarks to the Author):

Pich et al define mutational footprints in the genomes of AMLs and CH from patients treated with chemotherapeutics for solid cancers, providing new insight into the timing of bottlenecks and clonal expansions. The authors have thoroughly addressed reviewer concerns, by performing new analyses and simulations. Regarding the concerns of Reviewer 3 that the presence of clonal mutations with a platinum signature cannot be used to indicate the timing of the last bottleneck (or last common ancestor), the rebuttal of the authors is convincing and in my opinion their data do support their conclusions in this regard. That the majority of platinum related mutations are clonal is best explained by a selective sweep that occurred after drug exposure. They provide additional analyses restricted to clonal mutations to bolster this conclusion. I would also agree with the authors that “use of treatment mutations as barcodes to determine whether the clonal expansion occurs before or after the start of the exposure to the treatment is ... novel”, and adds significantly to our understanding of clonal dynamics in CH and AML evolution. They also use simulations to show that if the 5FU signature were present in the 3 analyzed tAMLs that they would have been able to detect it, addressing this concern raised by Reviewers 1 and 3. They also now show that the 5FU signature is detected in a subset of metastases dependent on certain variables (detailed below). Finally, issues of robustness of data are well addressed through their statistical analyses.

In all, this manuscripts provides novel insight into malignant evolution:

- 1) tAMLs from patients exposed to platinum-based chemotherapies (but not 5FU) exhibit a mutational footprint associated with these drugs. They conclude that “lack of the 5FU mutational footprint would indicate that cells that establish tAMLs were quiescent”, which is substantiated by simulations and via the detection of the 5FU signature in metastases from solid cancers.
- 2) They also analyzed chemo-footprints in the metastatic lesions from this cohort. It is interesting that treatment related mutations are only detected in metastases, and detection is more frequent in metastases that are distal to the primary tumor and with greater time from end of platinum therapy, both of which are plausibly proposed to reflect the increased odds of a bottleneck allowing fixation of particular mutations in one or a few cells (and thus mutations are detectable by WGS). Hence, they are able to infer the timing of the start of clonal outgrowth of the metastasis relative to treatment (if after treatment, therapy related mutations are more likely to be clonal).
- 3) Further analyses of the AML data show that the treatment likely predated the clonal expansion of the hematopoietic progenitor cells that founded the tAML. The loss of the linear relationship between the gain of HSC signature mutations (aging related) and age in tAMLs is very interesting, and could indeed reflect the compensatory proliferation during repopulation post-chemo. They used an unbiased method (IntOGen pipeline, based on recurrence, clustering and functional impact) to identify driver mutations in tAMLs, and observed increased driver mutations in TP53 as observed previously. They also observed more IDH1 mutations (in contrast to previous studies), and underrepresentation of NPM1 mutations. While drivers

identified are similar to those of other studies, the leveraging of the IntOGen pipeline adds robustness to these identifications. It would be interesting to know whether driver mutations identified in tAMLs or therapy-associated CH display the chemo signature. However, they would likely need to analyze (many) additional tAMLs to determine this. The available sample size is too small.

4) They characterized CH in the blood of the same cohort of patients with metastatic cancers. They observe significant overrepresentation of TP53 (consistent with these mutations conferring a higher risk of AML progress observed by others) and FLT3 (consistent with the occurrence of these mutations late in leukemia evolution), and underrepresentation of DNMT3a mutations. Interestingly, they were unable to detect the platinum mutation signature in CH from treated patients, although they were well powered to do so (based on new simulations). Thus, CH expansion likely occurred prior to the start of platinum treatment, with treatment favoring further expansion. Importantly, their demonstration that tAMLs do display such signatures indicates passage through a bottleneck during or after treatment (likely driven by additional driver mutations).

5) In all, leveraging chemo-related and other mutational signatures, this manuscript provides exciting new insight into clonal dynamics, particularly during CH and AML evolution, in response to chemo treatment. The new insight into the timing of bottlenecks and selective expansions enriches our understanding of malignant evolution in response to therapies.

We thank the reviewer for this detailed and appreciative summary of our work. We certainly share their enthusiasm for the results that we present.

Minor points:

1) The legend to Figure 4 is jumbled.

We thank the reviewer for pointing this out. We have corrected this error.

2) In the Discussion, I have trouble following this sentence: “This exposure appears as determinant, regardless of the phenotypic state of the exposed cell, to observe mutations related to treatments that directly damage the DNA, such as platinum-based drugs 27 .”

We agree with the reviewer that this sentence did not clearly convey the intended meaning. We have thus changed it as follows.

In other words, at least in the case of platinum-based drugs, which directly damage the DNA²⁷, the exposure to the drug produces a mutagenic effect of the same magnitude in both tumoral and healthy cells.

Signed: James DeGregori